# Oral Side Effects in Patients with Metastatic Renal Cell Carcinoma Receiving the Antiangiogenic Agent Pazopanib—Report of Three Cases

**DOI:** 10.3390/dj10120232

**Published:** 2022-12-08

**Authors:** Erofili Papadopoulou, Emmanouil Vardas, Styliani Tziveleka, Maria Georgaki, Maria Kouri, Konstantinos Katoumas, Evangelia Piperi, Nikolaos G. Nikitakis

**Affiliations:** 1Department of Oral Medicine and Pathology and Hospital Dentistry, School of Dentistry, NKUA, 11527 Athens, Greece; 2Department of Oral and Maxillofacial Surgery, School of Dentistry, NKUA, 11527 Athens, Greece

**Keywords:** pazopanib, angiogenesis inhibitors, gingival bleeding, jaw osteonecrosis, oral side effects, metastatic renal cell carcinoma

## Abstract

Pazopanib is a potent multi-kinase inhibitor that hinders angiogenesis and blocks tumor growth. It has been approved for the treatment of metastatic renal cell carcinoma (mRCC) and advanced soft tissue sarcoma. There is emerging evidence that bleeding is a common adverse effect of pazopanib and other targeted therapies in patients with mRCC. In addition, jaw osteonecrosis related to pazopanib was recently described in the literature. We report three cases of patients with mRCC who developed adverse oral events related to pazopanib. The first patient, treated with pazopanib as monotherapy, presented with gingival bleeding and oral burning sensation. The other two patients receiving pazopanib as monotherapy and pazopanib followed by sunitinib, respectively, presented complaining about mandibular pain; a diagnosis of medication-related osteonecrosis of the jaw (MRONJ) was rendered in both cases. Gingival bleeding and MRONJ may develop as oral side effects of pazopanib use. The cases presented here aim to alert and inform health care professionals about the risk of adverse oral events in patients with mRCC receiving the antiangiogenic agent pazopanib.

## 1. Introduction

In recent years, development of biological targeted therapies and immune checkpoint inhibitors has redefined the treatment for many cancers [1]. In order to avoid the severe adverse effects of conventional chemotherapy agents that are cytotoxic to all replicating cells, targeted therapies were designed to inhibit or minimize the development of cancer cells by attenuating, at the same time, the side effects of the therapy [2,3,4].

Inhibitors of angiogenesis are among the most common targeted therapies. Angiogenesis refers to the formation of new blood vessels, which facilitates the growth of the tumors as well as the spreading of the tumor though invasion of the vessels and entry into circulation. The main goal of antiangiogenic therapy is to inhibit pro-angiogenic molecules, such as Vascular Endothelial Growth Factor (VEGF), and, by doing so, to suppress the formation of new vessels [5,6]. Angiogenesis inhibitors are either monoclonal antibodies such as bevacizumab and aflibercept, or tyrosine kinase inhibitors (TKIs) such as sunitinib, sorafenib and pazopanib.

Several adverse effects have been associated with antiangiogenic therapy. Despite the fact that targeted therapies seem to display less or milder toxicities than conventional chemotherapy, several possible treatment complications have been reported including hypertension, thromboembolic events, cardiac dysfunctions, hypophosphatemia, thrombocytopenia, infections, chronic pain and skin conditions. Further, an increasing number of oral side effects of targeted therapies has been emerging, including medication related-osteonecrosis of the jaw (MRONJ) and oral mucositis, which may have a significant impact on patients’ quality of life [4,7,8,9,10,11]. Other possible complications of targeted therapies in the oral cavity include lichenoid lesions, xerostomia, dysgeusia and dysphagia [4,12].

MRONJ is a severe complication in cancer patients that manifests with many possible clinical signs and symptoms, including exposed bone, fistula, inflammation with or without suppuration, ulcers, soft and hard tissue necrosis, and edema. Pain, incapability of receiving food, paresthesia and trismus are complications arising due to jaw necrosis [13,14,15]. Since the first reports by Marx et al. in 2003 [13], MRONJ has been mainly associated with antiresorptive therapy (or with the so-called bone-targeting agents) that include bisphosphonates and denosumab [14,15]. However, there is an emerging incidence of MRONJ development in patients receiving antiresorptives in combination with antiangiogenic agents and/or mTOR inhibitors over the past few years [11]. More recently, MRONJ cases have been observed following administration of angiogenesis inhibitors without concurrent antiresorptives [11,14,15]. Among the antiangiogenic agents reported in the literature, bevacizumab and sunitinib were found to be the most common ones causing MRONJ [11].

Pazopanib is an oral small-molecule multi-kinase inhibitor that primarily inhibits VEGF receptor-1, -2 and -3, platelet endothelial growth factor receptor-α and -β, and the stem-cell factor receptor c-kit. Although pazopanib was developed as a therapeutic agent against various tumors, it is currently approved in many countries specifically for advanced soft-tissue sarcoma [11]. In addition, in October 2009, pazopanib was approved from FDA as first-line treatment for patients with advanced renal cell carcinoma (RCC) [12]. Reported side effects of the drug include diarrhea, nausea/vomiting, headache, loss of appetite, weight loss, altered sense of taste, temporary hair loss and/or change in hair or skin color. In addition, gingival bleeding related to pazopanib is very rare but a possible side effect in patients treated for RCC [8,10], while only a few cases report pazopanib or the similar drug axitinib as the main cause of MRONJ appearance [9,10,11,16].

The aim of the study is to present three cases of oral toxicities (one case of gingival bleeding and two cases of MRONJ) related to antiangiogenic treatment with pazopanib in patients with metastatic RCC (mRCC).

## 2. Case Reports

### 2.1. Case 1

A 64-year-old, non-smoking woman presented to the Department of Oral Medicine and Pathology and Hospital Dentistry with a chief complaint of oral bleeding and burning mouth sensation. Her medical history was significant for RCC diagnosed 5 months ago; 3 months later, she developed brain metastasis, for which she received radiation therapy to the brain (total dose of 30 Gy in 10 fractions). Since RCC diagnosis, she had been under chemotherapy with pazopanib (Votrient^®^, GlaxoSmithKline Inc., London, UK) daily as monotherapy. She was also receiving antihypertensive medication. Her complete blood counts were normal.

Clinical oral examination revealed local spontaneous gingival bleeding on the right anterior maxilla, as well as enlarged and erythematous gingiva associated with poor oral hygiene. According to her, she did not brush her teeth because of the bleeding and she was feeling nauseous due to the taste of blood; she also reported xerostomia (Figure 1a). For her oral condition, she had been receiving antifungal medication for 3 weeks prescribed by her medical oncologist without response.

Gingival bleeding was attributed to the antiangiogenic effects of pazopanib in combination with poor oral hygiene since she was not receiving any other medication with similar effects and her blood counts were normal. In collaboration with her medical oncologist, pazopanib administration was interrupted; antifungal medication was also discontinued since no signs of fungal infection were apparent. Oral hygiene instructions were delivered, including the use of soft toothbrush, interdental cleaning with suitable interdental brushes, and rinses with chlorhexidine 0.2% two times daily and warm chamomile mouthwash. Within one week, the patient reported regression of the bleeding and the burning sensation, although she still felt nauseous from the taste of the blood in her mouth (Figure 1b). One week later, no bleeding was noticed, and the patient reported further amelioration of the oral symptoms. The patient was referred for dental cleaning and, afterwards, the administration of pazopanib was recommenced. During her follow-up visits, no complaints were reported, oral health was satisfactory and there were no signs of gingival bleeding in oral examination.

### 2.2. Case 2

A 63-year-old non-smoking male patient was referred to our Clinic complaining about pain and pus discharge at the left retromolar area of the mandible. The patient was diagnosed approximately 1 year earlier with RCC (treated with nephrectomy) with concurrent lung and liver metastases. He had been receiving pazopanib daily as monotherapy for the last 9 months, but he had never received chemotherapy, any other anti-angiogenic or anti-resorptive medication. His medical history included diabetes mellitus and hypertension, both under medication. The left lower third molar had been extracted 8 months ago due to recurrent pericoronitis causing pain and swelling.

Oral clinical examination revealed exposed bone and significant pus discharge at the previous extraction site (Figure 2a); poor oral hygiene was also noticed. Panoramic radiograph showed a diffuse radiolucency in the area (Figure 2b). On the basis of the clinical and radiographic findings, a diagnosis of MRONJ, stage II according to the AAOMS staging system 2014 [14] associated with the use of pazopanib was rendered.

In collaboration with his medical oncologist and the referring oral and maxillofacial surgeon, a conservative approach was chosen, since his treatment with pazopanib could not be interrupted and the current medical status did not allow a surgical procedure. The patient was treated with local applications of medical O_3_ delivered in an oil suspension (28 applications at the area of exposed bone for 10 min each time, in an 8-month period of time), along with intermittent antibiotic treatment (antibiotics were administered when signs and symptoms of infection were present) and mouth rinses with chlorhexidine 0.2% two times daily. Oral hygiene instructions were also delivered. Nine months since the first visit, a bone sequestrum was formed (Figure 2c), and two months later, spontaneous exfoliation of the loose bone sequestrum was observed. During the last follow-up, 2 months later, there were neither complaints reported nor clinical signs of jaw osteonecrosis. A few months later, the patient passed away due to disease progression.

### 2.3. Case 3

A 56-year-old man was referred to our Clinic by his medical oncologist, complaining about pain in the right posterior mandibular retromolar area. A dental extraction of the right first lower molar had been performed 2 months earlier. Patient’s medical history was significant for hypothyroidism, hypertension and myocardial infarction, all under medication; additionally, he was diagnosed with RCC, approximately 3 years ago, concurrently with lung metastasis. He was initially treated with nephrectomy and 3 months later started receiving pazopanib daily (initiated approximately 2.5 years ago and administered for about 16 months as monotherapy), followed by sunitinib, once daily on a schedule of 4 weeks on treatment followed by 2 weeks off, which was interrupted by his medical oncologist due to jaw pain at the time of his referral.

Clinical intraoral examination revealed exposed bone at the extraction site accompanied by pus discharge (Figure 3a). Panoramic radiograph and cone beam computed tomography (CBCT) were performed and compared with an available panoramic radiograph before the dental extraction showing a periapical radiolucent lesion at the right first lower molar area (Figure 3b). Post-extraction radiographic evaluation revealed lack of bone healing at the extraction socket (Figure 3c), while CBCT demonstrated a diffuse osteolytic lesion encompassing a central hyperdense area at the extraction site (Figure 3d,e). Clinical and radiographic findings as well as history of administration of the antiangiogenic agents pazopanib and sunitinib led to a diagnosis of MRONJ stage II according to the AAOMS staging system 2014 [14].

Following consultation with the Department of Oral and Maxillofacial Surgery and since the patient was antiresorptive-naïve, meaning that he had never received antiresorptive agents such as bisphopshonates or denosumab, a conservative treatment was chosen, including intermittent antibiotic administration (antibiotics were administered when signs and symptoms of infection were present), analgesics and mouth rinses with chamomile and chlorhexidine 0.2% two times daily. Four months later, the patient was free of symptoms and clinical examination revealed full mucosal coverage at the post-extraction site (Figure 3f). Bone remodeling process was also confirmed radiographically (Figure 3g). No clinical signs or symptoms were reported during a 2-year follow-up period.

## 3. Discussion

Angiogenesis inhibitors, as well as inhibitors of molecular pathway of mammalian target of rapamycin (mTOR), are nowadays the mainstay of treatment for mRCC [10,17,18]. In fact, angiogenesis is upregulated via the production of VEGF in order to supply more oxygen due to hypoxia in cancer microenvironment. In cases of RCC, VEGF production is constantly increased. Antiangiogenic agents in mRCC patients target VEGF signaling, leading to inhibition of tumor-induced angiogenesis and, as a result, tumor growth suspension [17]. In late 2009, pazopanib obtained approval for patients with mRCC.

Even though angiogenesis inhibitors do not cause complications at a level similar to conventional chemotherapy, they still manifest various adverse events. The prevalence of oral toxicities of any grade in patients with mRCC is about 4% for pazopanib versus higher incidence for sunitinib and sorafenib (38% and 28%, respectively). In addition, regression of symptoms requires only dose reduction and not discontinuation of the drug [19]. It should be mentioned that oral adverse events in these patients are often underestimated, since patients rarely undergo an oral examination by an expert clinician. As a consequence, the reported prevalence of oral adverse events is low and diagnosis is frequently rendered at more advanced stages.

Among the oral adverse events of angiogenesis inhibitors, the most important ones, in particular from an oral surgery standpoint of view, are mucocutaneous bleeding and delayed wound healing. Hemorrhagic diathesis could be attributed to changes in vascular permeability caused by the antiangiogenic effects, and more often it manifests as nasal bleeding [1]. The incidence rate of bleeding events is up to 20–40% for bevacizumab versus 9–14% for pazopanib [1,20]. On the other hand, delayed wound healing could be explained since agents that inhibit VEGF affect vascular nourishment and re-epithelialization [21].

In the present study, our first case described gingival bleeding that could be, at least in part, attributed to the antiangiogenic effect of pazopanib. The blood counts were normal, pazopanib was the only administered medication that could affect hemostasis and oral hygiene was poor predisposing to gingivitis and local infections. Similar cases of gingival bleeding in patients receiving antiangiogenics have been published in the literature [21,22]. In these published cases, gingival bleeding was accompanied by necrotizing ulcerative gingivitis (NUG). Patients were treated conservatively with antibiotics, and, similar to our case, oral antiseptics and discontinuation of the angiogenesis inhibitor for a period of time. All signs and symptoms subsided and patients returned to their oncologic treatment.

MRONJ is a drug-related complication, observed in patients receiving mainly agents with antiresorptive and antiangiogenic properties. Initially, the complication was described in patients receiving antiresorptives, first bisphosphonates and later denosumab. Subsequently, several reports were published about MRONJ development in patients receiving bisphosphonates and concomitant bevacizumab or sunitinib. In these cases, a higher prevalence and worse clinical features were noticed, probably due to a cumulative toxicity profile [21].

In more recent years, there have been several reports about MRONJ in antiresorptive-naïve patients who had received only angiogenesis inhibitors. The majority of these cases were related to bevacizumab and sunitinib administration. Two of our cases demonstrated MRONJ development in antiresorptive naïve patients who received pazopanib as monotherapy (Case 2) or pazopanib followed by sunitinib (Case 3). There is another case published in the literature describing MRONJ development in a patient receiving pazopanib followed by everolimus [9].

The pathophysiology of MRONJ has not been fully clarified. Different hypotheses have been proposed to explain the pathogenesis of this complication. A quite dominant theory implicates the effect of the medication on macrophages, especially in antiresorptive-naïve patients as in our cases. Macrophages are derived from monocytes originating from bone marrow, have characteristics similar to osteoclasts and express VEGF receptors on their surface. Antiangiogenic agents cause reduction in macrophage numbers and activity by blocking VEGF pathway and, as a result, increase the risk for infections and tissue necrosis causing host defense impairment [7,10,22]. In addition, the interference in the natural angiogenesis, leading to reduced blood supply to the jaws and inhibition in bone repair, facilitates bacterial contamination of the exposed bone [8].

MRONJ in antiresorptive-naïve patients seems to have approximately the same predilection for gender, age, location (mandible) and dominant symptom (pain) as MRONJ in patients receiving antiresorptives and similar to our cases. A difference between the two conditions is the predominant cancer diagnosis, which in antiangiogenics-related MRONJ is advanced gastrointestinal cancer, followed by mRCC, whereas most cases of anti-resorptive-induced MRONJ are seen in breast cancer and multiple myeloma patients [23]. Furthermore, the mean time to MRONJ onset is shorter for angiogenesis inhibitors (38.8 weeks) than that of antiresorptives (136 weeks) [19]. On the other hand, the prognosis seems to be better in MRONJ associated with non-antiresorptive agents since resolution is more frequent (62% vs. 50%) and with shorter average time (6.75 vs. 8.2 months) [11]. These differences could be attributed to the shorter half-life and lower cumulative doses of angiogenesis inhibitors [11].

As far as treatment is concerned, conservative management is more commonly selected for MRONJ, according to the AAOMS position paper, and surgical approach is limited in persistent stage III that has failed to respond to conservative treatment [14,19]. In the cases presented here, Case 2 was treated conservatively, including local applications of ozone oil. According to the literature, ozone oil is produced by oxygen, which has a positive impingement on wound healing. It is reported that ozone oil is widely used as an antimicrobial agent and to promote wound healing due to its tissue repair properties [24]. As for Case 3, conservative treatment was also selected. Antibiotics, chlorhexidine and chamomile mouth rinses were used, the latter traditionally used as a treatment for many inflammatory diseases. Several literature reports refer to chamomile as a medical treatment with multiple applications in daily life; its antiaging, anticancer, anti-inflammatory, antioxidant, and antimicrobial properties establish it as a useful therapeutic approach in tissue repair process [25,26].

Pazopanib as well as other novel targeted therapies are widely used in clinical practice with remarkable effects, but they also provoke several adverse events. Health care professionals should be alerted to recognize initial and faded symptoms. A thorough oral and dental examination should be performed in all patients who receive or are about to receive angiogenesis inhibitors in order to maintain optimal oral health, reduce oral microbial load and achieve early diagnosis and prompt treatment of any complications, if needed.

## 4. Conclusions

Novel therapies such as pazopanib promote cancer regression and prognosis. However, oral adverse events may compromise the patient’s quality of life. Pazopanib, like other angiogenesis inhibitors, may be involved in the development of medication-related osteonecrosis of the jaws. In addition, gingival bleeding is another potential side effect of the medication, especially when combined with poor oral hygiene. Increased awareness of these possible adverse effects may facilitate their prevention, early diagnosis and adequate management.

## Figures and Tables

**Figure 1 dentistry-10-00232-f001:**
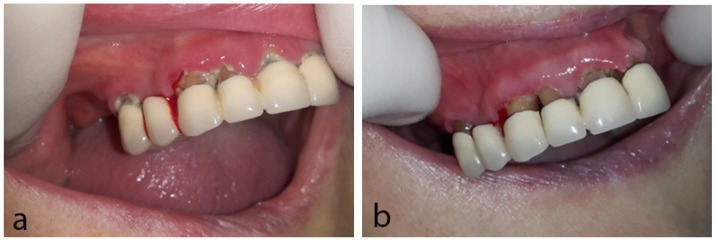
Case 1: (**a**) First clinical examination; spontaneous gingival bleeding and poor oral hygiene are noticed. (**b**) Follow-up at one week; amelioration of patient’s oral hygiene and remission of gingival bleeding.

**Figure 2 dentistry-10-00232-f002:**
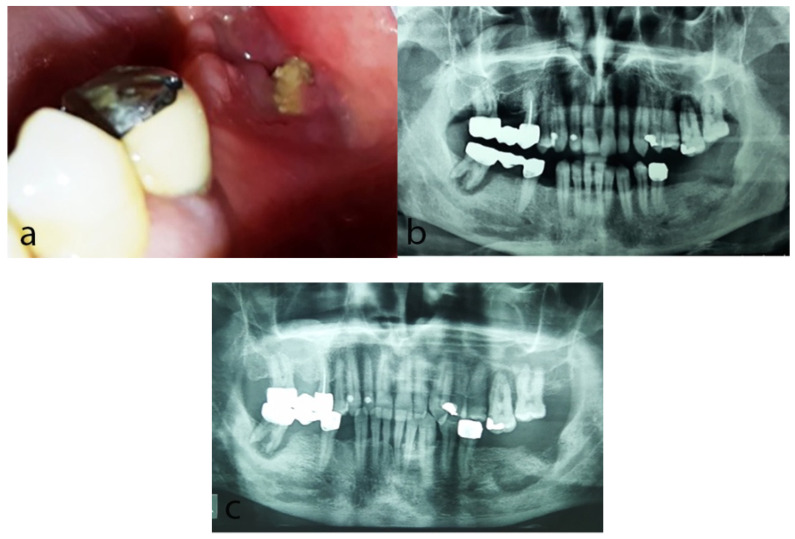
Case 2: (**a**) Clinical examination at the first visit revealing exposed necrotic bone in the left retromolar area at the site of a previous extraction before 8 months. (**b**) Panoramic radiograph at the first visit showing diffuse radiolucency in the left retromolar area. (**c**) Panoramic radiograph 9 months later showing persistence of the radiolucency and bone sequestrum formation in the same area.

**Figure 3 dentistry-10-00232-f003:**
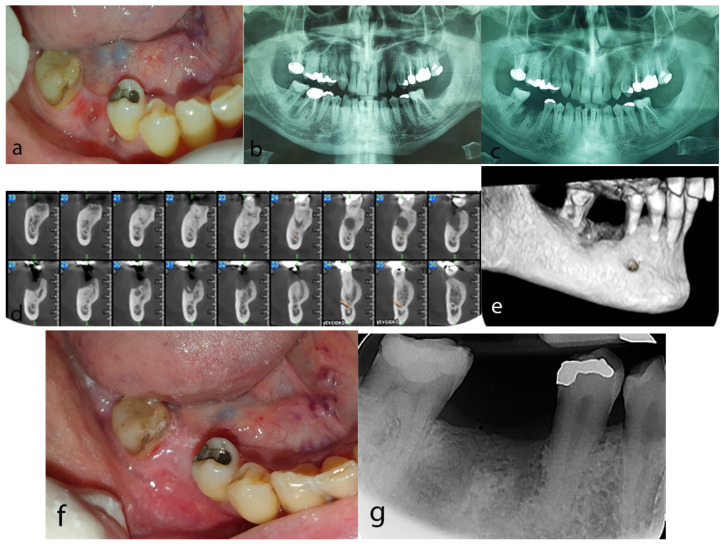
Case 3: (**a**) Exposed necrotic bone at the site of the extraction of the first right mandibular molar. (**b**) Panoramic radiograph before the extraction revealing radiolucent lesions around the apices of the right first mandibular molar. (**c**) Post-extraction panoramic radiograph showing phantom socket at the extraction site. (**d**,**e**) CBCT revealing a diffuse hypodense lesion containing a hyperdense area at the post-extraction site of the right mandible (**d**) cross sectional views, (**e**) 3D reconstruction). (**f**) Complete mucosal coverage at the post-extraction site after 4 months of conservative treatment. (**g**) Periapical radiograph demonstrating normal bone remodeling process after 4 months of conservative treatment.

## Data Availability

Department of Oral Medicine and Pathology and Hospital Dentistry.

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
