# Peer review of "Oral Side Effects in Patients with Metastatic Renal Cell Carcinoma Receiving the Antiangiogenic Agent Pazopanib—Report of Three Cases"

_dentistry, 2022, doi:10.3390/dj10120232_

Round 1
Reviewer 1 Report
Dear authors,
the title must be improved - it is doubled
Why are the 3 cases relevant to the literature?
keywords should be reconsidered - MEsh terms are recommended
Why is the aim to report those cases?
Explain antiresorptive-naïve
Treatment of the cases should be better pointed
Clinical significance of the paper should be described in the context of pazopanib use
Figure legends must not be presented at the end again
Images should be presented enlarged - they are barely distinguable
For case 1 panoramic radiography should be presented
What is the differne between figure 3 a and f?
What are the limitantions of the paper?
What is the originality?
Author Response
Reviewer #1
Point 1: The title must be improved-it is doubled.
Response 1: Thank you for this comment. The title is not doubled, it is the main title followed by the short title; we have now tried to clarify this by separating the two titles. Also, we improved the main title of the manuscript by removing “review of literature”.
Point 2: Why are the 3 cases relevant to the literature?
Response 2: Thank you for this comment. Angiogenesis inhibitors, like pazopanib, have been previously implicated for their role in the development of medication-related osteonecrosis; in some cases, a combination of angiogenesis inhibitors and antiresorptive agents was administered, while, in other cases, jaw osteonecrosis was associated with the use of angiogenesis inhibitors only. Since these medications, especially pazopanib as in our cases, are commonly used, both dentists and medical oncologists should be aware of this potential adverse event and two of the cases presented here highlight this probability. Moreover, bleeding is another significant adverse event of angiogenesis inhibitors and for pazopanib it has only been reported in clinical trials; therefore, the case presented here is unique in emphasizing the potential for oral bleeding development in patients receiving pazopanib.
Point 3: Keywords should be reconsidered-Mesh terms are recommended
Response 3: Thank you for this useful comment. Keywords have been replaced in the section of Abstract-keywords (Page 1).
Point 4: Why is the aim to report those cases?
Response 4: Thank you for this comment. There are only a few articles in the literature concerning the oral side effects of pazopanib. Dentists and medical oncologists should be aware of the adverse effects of the drug as to inform their patients and apply all the necessary preventive measures to avoid them.
Point 5: Explain antiresorptive-naïve.
Response 5: Thank you very much for this comment. Antiresorptive-naïve means that the patient has never received antiresorptive agents such as bisphosphonates or denosumab. This has now been clarified in the Case Report section, case 3 (page 5, 1st paragraph, lines 183-184).
Point 6: Treatment of the cases should be better pointed.
Response 6: Thank you for this comment. Treatments are now pointed in more details in the Case Report section, Case 1 (page 3, 1st paragraph, lines 107-108), Case 2 (page 3, 4th paragraph, lines 138-141) and Case 3 (page 5, 1st paragraph, lines 185-187).
Point 7: Clinical significance of the paper should be described in the context of pazopanib use.
Response 7: Thank you very much for this comment. The clinical significance is that patients who are about to receive or are already receiving pazopanib, should undergo thorough oral and dental examination, use oral hygiene measures and have regular dental follow-ups in order to avoid or diagnose as early as possible any oral adverse events related to this drug. The clinical significance has now been mentioned in the Discussion section (page 7, 2nd paragraph).
Point 8: Figure legends must not be presented at the end again.
Response 8: Thank you for this comment. The figure legends have been now removed from the end of the manuscript.
Point 9: Images should be presented enlarged- they are barely distinguishable.
Response 9: Thank you for this comment. We enlarged all figures as much as it was possible.
Point 10: For case 1 panoramic radiography should be presented.
Response 10: Thank you for this comment. Unfortunately, there is no panoramic radiograph available for case 1. However, we think that the main relevant clinical finding of this case, i.e. gingival bleeding, is illustrated sufficiently in the clinical photos.
Point 11: What is the differne between figure 3a and f?
Response 11: Thank you for this comment. Figure 3a presents two small lesions of exposed bone at the site of the extraction of the first right mandibular molar whereas Figure 3f presents the same site after 4 months of conservative treatment where complete mucosal coverage is achieved.
Point 12: What are the limitations of the paper?
Response 12: Thank you very much for this comment. The limitations of the paper are the relatively low number of reported cases, which however highlights the need for awareness of this condition, which could ideally result in publication of more relevant cases. Further, there was no long-term follow-up of the three patients.
Point 13: What is the originality?
Response 13: Thank you for this comment. To our knowledge, this is the first case report of gingival bleeding related to pazopanib. So far, bleeding (nasal or gingival) has been reported as an adverse event of the drug only in clinical trials. In addition, there have been only two case reports concerning medication-related osteonecrosis of the jaws related to pazopanib.
Reviewer 2 Report
Thank you for sending us your article and giving us the chance to consider your work. Your article was read . I enjoyed reading your article and are pleased to offer of publication ,it was well written and ready to be published.
Author Response
Reviewer #2
Thank you for sending us your article and giving us the chance to consider your work. Your article was read. I enjoyed reading your article and are pleased to offer of publication, it was well written and ready to be published.
Response: We sincerely appreciate the reviewer’s very positive evaluation of our work and we would like to thank the reviewer for appreciating the overall interest of our manuscript.
Reviewer 3 Report
Thank you for submitting: "Oral side effects in patients with metastatic renal cell carcinoma receiving the antiangiogenic agent pazopanib. Report of three cases and literature review." The aim of the study is to present three cases of oral toxicities (one case of gingival bleeding and two cases of MRONJ) related to antiangiogenic treatment with pazopanib, in patients with metastatic RCC (mRCC).
I find it an interesting article, which can provide good information for the use of pazopanib. Only some minor concerns:
what camera and what flash were used to take the pictures?
What radiographic equipment was used?
What is the registration number of the ethics committee? also attach it
Author Response
Reviewer #3
Thank you for submitting: ‘’Oral side effects in patients with metastatic renal cell carcinoma receiving the antiangiogenic agent pazopanib. Report of three cases and literature review.’’ The aim of the study is to present three cases of oral toxicities (one case of gingival bleeding and two cases of MRONJ) related to antiangiogenic treatment with pazopanib, in patients with metastatic RCC (mRCC). I find it an interesting article, which can provide good information for the use of pazopanib. Only some minor concerns:
Response: We really appreciate reviewer’s overall positive assessment of our
manuscript.
Point 1: What camera and what flash were used to take the pictures?
Response 1: Thank you for this comment. We use a Canon camera of high quality, which complies with the Canadian ICES-003 Class B specifications.
Point 2: What radiographic equipment was used?
Response 2: Thank you for this comment. We use high quality, standard equipment. For panoramic radiographs a Planmeca device is used and for the periapical radiographs, the Gendex oralix AC.
Point 3: What is the registration number of the ethics committee? Also attach it.
Response 3: Thank you for this comment. The study protocol was approved by the Ethical Committee of School of Dentistry of National and Kapodistrian University of Athens and the reference number is 543/4-11-2022.
Reviewer 4 Report
Consider shortening the title; Maybe removing the review of literature since the manuscript is more of a case series report.
Figures 2 and 3 should be placed right beneath the text, aligned to the left.
The conclusion section should be reduced and report findings. to the reported strictly related.
Author Response
Reviewer #4
Point 1: Consider shortening the title; Maybe removing the review of literature since the manuscript is more of a case series report.
Response 1: Thank you very much for this comment. Indeed, we restricted the title by removing the review of the literature, as you suggested.
Point 2: Figures 2 and 3 should be placed right beneath the text, aligned to the left.
Response 2: Thank you for this comment. We have now placed Figures 2, 3a to 3e and 3f and g right beneath the text, aligned to the left, according to the text.
Point 3: The conclusion section should be reduced and report findings. To the reported strictly related.
Response 3: Thank you for this comment. The conclusion was modified focusing now on the reported findings, as suggested (page 7).
Round 2
Reviewer 1 Report
The manuscript has been improved
Reviewer 3 Report
The authors have followed the changes suggested and have greatly improved the article.
Therefore, in my opinion, this scientific article meets the necessary criteria to be published in
present form.